# Artificial Intelligence in Decrypting Cytoprotective Activity under Oxidative Stress from Molecular Structure

**DOI:** 10.3390/ijms241411349

**Published:** 2023-07-12

**Authors:** Damian Nowak, Karolina Babijczuk, La Ode Irman Jaya, Rafał Adam Bachorz, Lucyna Mrówczyńska, Beata Jasiewicz, Marcin Hoffmann

**Affiliations:** 1Department of Quantum Chemistry, Faculty of Chemistry, Adam Mickiewicz University in Poznan, Uniwersytetu Poznanskiego 8, 61-614 Poznan, Poland; 2Department of Bioactive Products, Faculty of Chemistry, Adam Mickiewicz University in Poznan, Uniwersytetu Poznanskiego 8, 61-614 Poznan, Poland; 3Department of Cell Biology, Faculty of Biology, Adam Mickiewicz University in Poznan, Uniwersytetu Poznanskiego 6, 61-614 Poznan, Poland; 4Institute of Medical Biology of Polish Academy of Sciences, Lodowa 106, 93-232 Lodz, Poland

**Keywords:** artificial intelligence, machine learning, antioxidant properties, indole derivatives

## Abstract

Artificial intelligence (AI) is widely explored nowadays, and it gives opportunities to enhance classical approaches in QSAR studies. The aim of this study was to investigate the cytoprotective activity parameter under oxidative stress conditions for indole-based structures, with the ultimate goal of developing AI models capable of predicting cytoprotective activity and generating novel indole-based compounds. We propose a new AI system capable of suggesting new chemical structures based on some known cytoprotective activity. Cytoprotective activity prediction models, employing algorithms such as random forest, decision tree, support vector machines, K-nearest neighbors, and multiple linear regression, were built, and the best (based on quality measurements) was used to make predictions. Finally, the experimental evaluation of the computational results was undertaken in vitro. The proposed methodology resulted in the creation of a library of new indole-based compounds with assigned cytoprotective activity. The other outcome of this study was the development of a validated predictive model capable of estimating cytoprotective activity to a certain extent using molecular structure as input, supported by experimental confirmation.

## 1. Introduction

Machine learning algorithms are currently being used to accelerate the discovery of novel chemical structures with specific properties [1,2,3,4,5]. These approaches require some preliminary data, which consist of known structures for substances with determined interesting target features from which the mathematical models can learn. The results can be utilized to create new potentially active chemicals. It is possible to do so by employing a computer-friendly linear representation of a molecule, such as SMILES [6]. The methodology created by us previously [1] lets us construct a library of novel compounds from existing structures. The desired feature (cytoprotective activity against oxidative damage) can be assigned to previously untested structures [7].

The structures under consideration here were indole derivatives. They are considered potentially beneficial cytoprotective, antioxidant, antibacterial, antiviral, and fungicidal compounds [7,8,9,10,11]. The cytoprotection of indole based-derivatives is associated with the inhibition of hemolysis induced by free radicals [7,11,12]. The cytoprotective activity of biocompatible compounds is based on the scavenging of free radicals in the cellular environment and/or their incorporation into the cell membrane, thus stabilising molecular structure. The cytoprotective activity of bioactive compounds is defined as the inhibition of hemolysis induced by free radicals and measured as a percentage (%). An excessive amount of free radicals can lead to lipid peroxidation in the erythrocyte cell membrane, resulting in changes in its molecular structure and ultimately leading to oxidative hemolysis. Human erythrocytes, as carriers of oxygen, are more susceptible to oxidative damage compared to other cells due to the oxygen-transporting hemoglobin and high content of polyunsaturated fatty acid in their cell membranes. Therefore, bioactive compounds with antioxidant properties play a crucial role in protecting cells from damage caused by oxidative stress [7,11,12].

Oxidative stress is defined as an imbalance between the production of reactive oxygen species (ROS) and the antioxidant system’s efficiency. Overproduction of ROS has been linked to cancer, cardiovascular, neurological, and autoimmune diseases [13]. ROS can cause lipid and protein peroxidation and nucleic acid damage in a concentration- and time-dependent manner. Recently, exogenous antioxidants have garnered significant attention due to their ability to prevent oxidative damage in cells. Indole derivatives, in particular, have demonstrated notable antioxidant and cytoprotective properties [11,12,14]. The electron-rich aromatic ring structure of indole antioxidants allows them to operate as electron donors for the generation of cationic radicals or through the addition of electrophilic radicals at the C-3 position of the indole ring [15].

Many indole-based structures have been approved for use as pharmaceuticals. For example, indomethacin is regarded as one of the most promising analgesic and anti-inflammatory drugs [16]. Pindolol has been used to treat hypertension since 1982 [17]. Indapamide is used to treat heart failure and hypertension [18], and delavirdine is used to treat HIV-1 [19]. These data confirm that indole derivatives have numerous practical applications [20].

Our study can be divided into three sections: two of which are theoretical and the third is strictly experimental. The primary objective of this project was to propose novel indole derivatives by utilizing our previously developed AI prediction methodology and building upon known structures [1]. The secondary goal was to create a predictive model that can provide us with feedback on potential cytoprotective action. The last objective was the experimental verification of previously untested structures’ predicted activity to determine whether they are indeed active. Based on the limited number of measurements with known target values, the correlation between predicted and measured cytoprotective activity was expected to be useful but not completely accurate. As only a few data points were available, the model for predicting cytoprotective activity was kept simple.

A comparison of the newly developed structures with the starting ones was performed. It can inform us about the chemical space encompassing novel and initial structures. A search was conducted to determine whether the newly produced structures could be found in the PubChem database [21].

## 2. Results and Discussion

### 2.1. New Structure Generation

Based on 44 initial structures (Appendix A), we generated 134,373 unique chemical representations using the SELFIES notation. They all described initial structures. This indicated that each of the initial structures could be presented in a variety of ways (Appendix A). Then, throughout the machine learning process, the neural network constructed (Appendix A) demonstrated the ability to learn how to recreate chemical structures. Figure 1 shows the decreasing loss value, indicating that the model performed more accurately as the learning time increased (Appendix A).

Based on 44 starting structures (Appendix A), we were able to generate 891 distinct structures (Appendix A), which were distinct from the initial structures yet somehow comparable in the sense of Tanimoto similarity [22] (Appendix A).

Some of them are depicted below (Table 1). This table shows the potential generative application of the neural network employed [1]. The table contains three selected structures from the initial dataset, along with the three most similar generated structures for each of them based on Tanimoto similarity. Table 1 contains information about the exemplary initial structures with known cytoprotective activity (%), their SMILES [6] codes, and selected highly similar newly generated structures with marked predicted cytoprotective activity (%) (Appendix A). It can be seen that small changes in structure may lead to increased cytoprotective activity (Table 1, third column) or decreased cytoprotective activity (Table 1, first column).

### 2.2. Cytoprotective Activity Prediction

To predict the cytoprotective activity parameter, five alternative approaches can be used. Table 2 records their performance.

The table shows:The method for predictive model construction based on the approaches listed in Section 3.3;The correlation threshold—meaning the correlation between molecular descriptors and measured cytoprotective activity (target feature);The number of features (molecular descriptors)—this is closely related to the correlation threshold; the higher the correlation threshold, the fewer features can be used to form a model;Standardization of features—when set to “True”, features are standardized; when set to “False”, features are not standardized;The training R score indicates how well a model predicts cytoprotective activity (the correlation coefficient between real and predicted values) based on the data seen during training;The testing R score indicates how well a model predicts cytoprotective activity (the correlation coefficient between real and predicted values) using data that were not present during training;The mean squared error (MSE) displays both the estimator’s bias (accuracy), which is how much its expected value systematically differs from the true value, and the estimator’s variance (precision), which shows how much it fluctuates around its expected value owing to sampling variability [25];The mean absolute error (MAE) indicates the average variation between the significant values in the dataset and the projected values in the same dataset [26].

The decision tree (DT) [27] (Appendix A) and random forest (RF) [28] (Appendix A) algorithms were the best-performing techniques according to Table 2. The latter is constructed from the former and consists of numerous decision trees. The prediction is based on the average from multiple decision trees. These models appear to be reliable for predicting cytoprotective activity.

Another method demonstrated that a small number of data points is insufficient to develop a high-performance K-nearest neighbors (KNN) algorithm [29] (Appendix A), despite the fact that it had a very high testing R score. It is possible that this was observed due to the small number of testing points. However, the training correlation coefficient was insufficient. This model was not appropriate for use with the given dataset.

Multiple linear regression (MLR) [30] (Appendix A) requires the use of up to nine features to build an average model. Given that we only had 44 data points, this is an excessive number of features. The more features are used, the more likely the model is to overfit [31] to the training data. As a result, the model will be less applicable due to likely underfitting [31] to the testing data. Inappropriate generalizations may be formed. This model was not applicable to the given dataset.

Support vector regression (SVR) [32] (Appendix A) performed badly since it requires as many as 46 features to obtain a training correlation coefficient of 80% and it cannot work with previously unseen data due to underfitting to the testing set. This model was not suitable to be employed with the given dataset.

**Table 2 ijms-24-11349-t002:** Predictive model approaches investigated for the prediction of cytoprotective activity (Appendix A).

Method	Correlation Threshold	Number of Features	Standardization of Features	Training R Score	Testing R Score	Mean Squared Error	Mean Absolute Error
DT [27]	0.39	2	False	0.91	0.80	142.31	6.77
RF-12 [28]	0.39	2	False	0.85	0.75	208.96	10.54
RF-6 [28]	0.39	2	False	0.84	0.73	226.73	10.85
KNN [29]	0.39	2	True	0.50	0.93	975.34	20.82
MLR [30]	0.34	9	False	0.55	0.37	530.98	16.57
SVR [32]	0.25	46	True	0.80	0.00	1.28 × $10^{12}$	1.84 × $10^{5}$

The “best”-performing model was chosen to make predictions—a random forest model with six estimators (RF-6). It had slightly lower quality measurements than the RF model with 12 estimators (RF-12), but the 6-estimator model was less complex than the 12-estimator model (Table 2). Figure 2 depicts the workflow with the various numbers of estimators. The best-performing model was defined as that with the highest training and testing R scores and using the fewest features. The random forest model was chosen because the decision tree described in Table 2 had a relatively restricted range of predicted values—only 23 possibilities for cytoprotective activity. Despite having the best R score for both training and testing, it was not chosen as the final model.

The final random forest model employed six decision trees and presented the mean of their outputs. In the case of a single decision tree, the universe of possible cytoprotective activity predictions is narrower. The model also performed quite well.

The selected model worked in the manner shown in Appendix A. This explains how the model made decisions about the assignment of cytoprotective activity based on molecular descriptors. Appendix A displays only one of the six decision trees involved in the final prediction. The two descriptors can be found in the model constructors: ETA_dEpsilon_D [33,34,35], which describes a measure for the contributions of hydrogen bond donor atoms, and nHBDon [33,36], which describes the number of hydrogen bond donors. Both of them are related to hydrogen bond donor numbers, and this information can be suitable for capturing the relevant information for cytoprotective activity under oxidative stress conditions.

### 2.3. Structures for the Experimental Verification

The SYBA score, helpful in assessing if a molecule is easy to synthesize, [37], was used in the selection process (Appendix A). It allowed us to minimize the number of distinct structures created from 891 to 213. For initial structures, the lowest SYBA score was 42.29. This meant that the SYBA scores for the 213 structures were greater than or equal to 42.29. Some of the structures chosen were subject to biological testing to see if the projected cytoprotective activity could be confirmed (Appendix A).

The structures in Appendix A were chosen for experimental verification. Some of the indole-based compounds were biologically examined. These chemicals were chosen to test the prediction capacities of the cytoprotective activity random forest model. Future studies will focus on the newly established library of compounds with predicted cytoprotective activity.

### 2.4. Additional Analysis of AI-Generated Structures

Appendix A contains the generated structures already present in the PubChem database. Their CIDs were collected along with generic PubChem SMILES. A total of 61 of the 891 generated structures were found in the PubChem database.

Appendix A contains the histograms associated with Tanimoto similarity. They demonstrate that the SYBA score selection led to the sieving out of less similar structures while preserving those with a higher similarity. This observation is depicted in Figure 3.

For the sake of curiosity, the chemical space from the molecular fingerprints was created. It was employed in the t-SNE dimensionality reduction [38]. It gave us information about the similarity between the initial and new structures (Appendix A).

In Figure 4, the chemical space for all the generated structures (891 (Appendix A)) and initial structures (44) is presented. It shows that the generated structures were partially similar to the starting ones. However, some of the newly generated molecules were in a different chemical space than the starting ones (Appendix A).

After applying the SYBA algorithm, the chemical space was created for the 213 selected structures and 44 initial structures. This can be viewed in Figure 5. It can be observed that the application of the selection step resulted in the preservation of structures that were more similar to the starting ones. It should also be mentioned that the chemical space for all the generated structures had the following coordinates: (TSNE_C1) −40 to 40 and (TSNE_C2) −30 to 40. The new chemical space had the following coordinates: (TSNE_C1) −20 to 15 and (TSNE_C2) −20 to 15. This strictly shows that the similarity increased (Appendix A).

### 2.5. Experimental Results

The experimental data are presented in the Appendix A). The experimental data showed that some indole-based structures (compounds three and four) had higher cytoprotective activity and others (compounds one and two) had lower cytoprotective activity than predicted by the RF model (Table 3). The tendency of the predicted cytoprotective activity was reversed, meaning that structures that were predicted to be very active were less active and the structures predicted to have medium activity had medium activity in the experiment. This was observed probably due to the distribution of the cytoprotective activity in the training dataset (Appendix A), which was skewed to the higher cytoprotective activities. This suggests the need to use more data points with structures that are less active and to rebuild the RF model with them. Moreover, our model used a very limited number of molecular descriptors: only two were employed in the prediction formation.

The methodology is ready to be employed with bigger datasets in the future. It may be valuable to re-predict cytoprotective activities for AI-generated structures with the enhanced predictive model.

The results presented in Table 3 were unified with the cytoprotective activities used as the training dataset. This was carried out by using the known cytoprotective activity of the reference compound Trolox [7,11] (concentration: 0.025 mg/mL) [11]. In comparison, in this experimental verification, the other concentration was used (0.1 mg/mL). The results were scaled up for the higher concentration. The agreement between the predicted cytoprotective activity under oxidative stress and the in vitro measured activity was significant.

According to the results obtained in the in vitro evaluation using human erythrocytes as a cell model (Appendix A)), the tested compounds showed potential for biomedical applications. All compounds tested showed the ability to inhibit free radical-induced hemolysis. Compound three showed the highest cytoprotective activity, providing membrane protection against oxidative damage.

## 3. Materials and Methods

The names of files are stored in the Appendix A).

### 3.1. Initial Structures

The origin structures were structures with known cytoprotective properties as determined in prior investigations (Appendix A) [7,11,12,14,39]. They served as the foundation for the development of novel structures, as well as the development of cytoprotective activity prediction models.

These structures were then recorded with different SMILES [6] representations for each one. We created 134,373 unique chemical representations using the SELFIES [40] notation based on 44 initial structures’ SMILES. While neural networks require a large amount of data for training, these structures represent 44 basic structures in various ways. This was accomplished through the use of the RDKit library [41] and a transition from the RDKit molecular entity to SMILES representation. As the neural network took advantage of SELFIES rather than SMILES, translation was required from one form to another (Appendix A). The structures are stored in Appendix A.

### 3.2. New Structure Generation

The new structures were proposed using the neural network [1]. The neural network used here is detailed in great depth in Appendix A. As neural networks are intended to represent data mathematically, our linear representation of structures had to be vectorized. Vectorization is the process of converting a computer-unreadable representation of data through mathematical processing into computer-readable objects known as mathematical vectors [42]. The chemical structures were sent into the neural network as vectorized representations of SELFIES. This allowed us to successfully proceed with the data and learn the rules of chemical structure formation using the neural network. The neural network’s major goal was to learn how to appropriately recreate chemical structures provided during training.

The 134,373 distinct SELFIES (Appendix A) that comprised our initial structures were divided into training (120,935) and validation (13,438) sets. The validation set informed us about how well the model reconstructed unknown chemical structures. The loss parameter was calculated using the categorical cross-entropy function [43].

During the vectorization process, two new letters were added: “!” for the beginning of a structure and “E” for the end of a structure. These characters were vectorized as well. This allowed the vectors to be the same length for all training structures (Appendix A).

The model was then prompted to predict new structures from the latent space, and noise was introduced. While the training data used the representations of just 44 structures and we sought to develop new structures based on that limited number, the noise let us create more new structures. Twenty predictions were made for each of them. The new structures were constructed based on how likely it was that a specific atom would be present at a given place. When the final character was encountered, the new structure was completed [1] (Appendix A). The prediction results are stored in Appendix A.

### 3.3. Cytoprotective Activity Prediction

The cytoprotective activity prediction was carried out for this small dataset (44 points) with the following method. We used random samplings when assigning a given structure to the “train” and “test” datasets. The “test” set contained a low-activity structure, a medium-activity one, and a high-activity one, as the whole dataset was very small in this study. The use of regression predictive models to predict cytoprotective activity [%] was assumed. The following approaches were tested based on the assumption: the decision tree (DT) model [27] (Appendix A), the K-nearest neighbors (KNN) model [29] (Appendix A), the random forest (RF) model [28] (Appendix A), the SVR model [32] (Appendix A), and the multiple linear regression (MLR) model [30] (Appendix A).

The Mordred library [44], which is an RDKit implementation, was used to calculate the molecular descriptors [45]. Then, for each model, the correlation coefficient between the target value and the molecular descriptors was calculated. A standardization parameter that could be true or false was also used and concerned the standardization of molecular descriptor features (Appendix A).

Each model was tested for various correlation thresholds, as well as with and without standardization. The lower the number of features utilized to form the prediction model, the higher the correlation threshold was (Appendix A). As we wanted to obtain feedback about the generalization of the cytoprotective activity predictive model, some data points were used as test points. These points were omitted during training and were used later to obtain information about our model’s performance. They served as a reference for our model, allowing us to check if it performed well with previously unseen data (Appendix A).

The last thing was the creation of the final model based on the quantitative parameter R (correlation coefficient [46]); the higher the value, the better our model performed. Based on the training and testing R values, one model was chosen. The model was used to predict the cytoprotective activity [%] of the newly generated structures, and the outputs were recorded.

### 3.4. Structures for the Experimental Verification

As many structures were produced, the number of results had to be reduced. This was undertaken using the SYBA algorithm [37]. As a result of the algorithm, the SYBA score was produced; the higher the SYBA score, the easier it may be to synthesize the molecule. The score was calculated for the initial structures, and the lowest obtained value was used as a threshold for the newly generated structures. This stage also gave predicted cytoprotective activity for each of the structures chosen (Appendix A). Appendix A contains the results of the SYBA selection. After this, the size of the library of new structures was decreased.

The potential cytoprotective activity of previously untested structures was predicted using the random forest algorithm [28]. The prediction is shown at the end of Appendix A, and the results are in Appendix A. The experimental confirmation of the investigated structures is stored in the Appendix A), where their synthesis and the spectroscopic analysis are described.

### 3.5. Additional Analysis of AI-Generated Structures

We searched for the generated structures in the PubChem database with the application of the PubChemPy pythonic library [21]. It gave information about whether the structure generated could be found in the PubChem database (Appendix A). The results of the search are stored in Appendix A.

The similarity calculation was performed using Tanimoto similarity parameter [47] (Appendix A). It gave information about whether two structures were similar in the sense of molecular fingerprint similarity. A molecular fingerprint is a simplified depiction of some characteristics of a specific molecule. It is a concise, binary digit-based representation of a chemical structure. The RDKit fingerprint was utilized here, which is yet another implementation of a daylight-like fingerprint [48]. With the molecular fingerprint representation, the similarity of two species can be simply calculated [22,49]. The Tanimoto similarity coefficient, in this instance, involved two fingerprints and showed the similarity between the molecules. In the corresponding bit representations, the value 1 reflected identical molecules, whereas the value 0 indicated that no shared components existed. Appendix A resulted in the generation of Appendix A, which also contains information about the newly generated structures discovered in the PubChem database (Appendix A). Tanimoto similarity histograms are shown in Appendix A. Tanimoto similarity was present between all created structures, SYBA-selected structures, and beginning structures. Appendix A is another Tanimoto similarity Excel file but solely for the structures that were selected (Appendix A).

Based on t-distributed stochastic neighbor embedding (t-SNE) analysis [38]—a dimensionality reduction algorithm—the chemical space of the created structures was compared to the initial structures. This approach allowed us to separate data that could not be divided by a straight line—hence the name “nonlinear dimension reduction”. It enabled us to comprehend high-dimensional information and transfer it into a low-dimensional space. It enabled us to reduce the size of each molecule’s molecular fingerprint and present further similarities between the new structures and the starting structures (Appendix A).

### 3.6. Experiment Description

The synthesis of each of the tested compounds is shown in the Appendix A). The spectroscopic data allowed us to confirm that the tested structures had been properly synthesized(^1^H NMR, ^13^C NMR, EI-MS, and IR). The spectra for the tested compounds **1**–**4** can also be viewed there (Appendix A).

The following parameters were experimentally tested:Hemolytic activity [50];Cytoprotective activity under oxidative stress conditions [7,11,12,14,39].

Detailed descriptions and the results of each assay are provided in the Appendix A).

Hemolytic activity was determined for all compounds tested to assess their sublytic concentration. The cytoprotective activity of these compounds at a sublytic concentration was then estimated. The results of this assay are displayed in Appendix A.

The cytoprotective activity was measured for the concentration of 0.01 mg/mL. The results of the cytoprotective activity investigations for compounds **1**–**4** are presented in Appendix A.

## 4. Conclusions

The proposed methodology let us create an AI model that has some predictive capabilities related to cytoprotective activity. More importantly, we showed that the AI model could predict novel, chemically meaningful structures with beneficial biological properties. Our methodology shown here can be used in other quantitative structure–activity relationship (QSAR) studies. In this study, we evaluated various AI approaches: the RF, DT, MLR, SVR, and KNN models. Therefore, we could select the best solution for predicting the cytoprotective activity of the compounds tested under oxidative stress conditions. The created RF model performed quite well with training and testing data, although the distribution of training data points was skewed towards higher activities. Surprisingly, it was found that some capabilities for the recognition of cytoprotective activity patterns were gathered by the RF model. The experimental study supported the AI model’s ability to predict cytoprotective properties under oxidative stress conditions to a certain extent and inform experimenters of more suitable chemical substituents. Further efforts may be directed towards gathering a larger dataset, which would let us use more molecular descriptors to build an upgraded AI model. The model can also be retrained with the AI-generated structures that should have been previously synthesized and experimentally verified.

This model has not been used with structures other than indole-based compounds. As the machine learning model possesses more interpolation than extrapolation capabilities, it can achieve much higher certainty in the results for more similar structures. If one wants to predict cytoprotective activity under oxidative stress for totally different compounds, it would be less certain. This means that we can be more sure of the model’s prediction if the structure of the object of our consideration is closer in chemical space to the training data. This is the limitation of the model. Bias in AI algorithms skews results in favor of or against an idea. It is a systematic error caused by incorrect assumptions in the AI learning process. In this manner, it can affect the construction of an AI model.

## Figures and Tables

**Figure 1 ijms-24-11349-f001:**
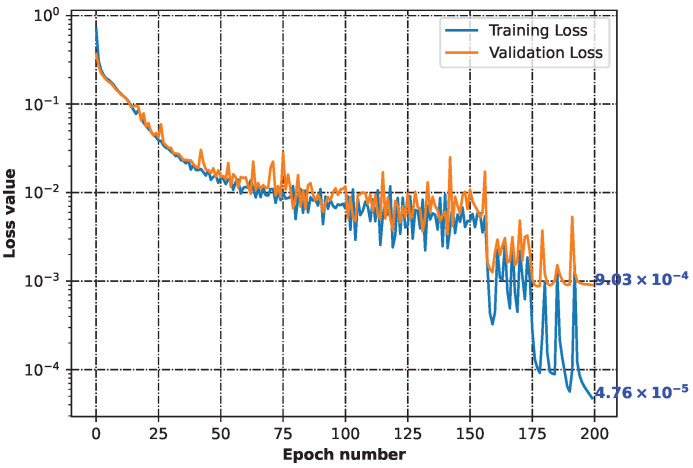
Training and validation losses of the neural network minimization. Both parameters dropping indicates that the model was learning how to generalize from the input.

**Figure 2 ijms-24-11349-f002:**
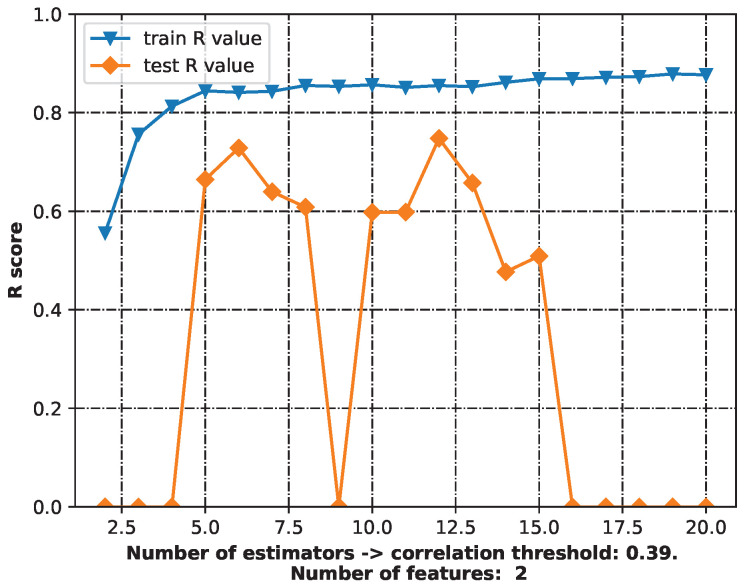
The workflow for the best-performing model.

**Figure 3 ijms-24-11349-f003:**
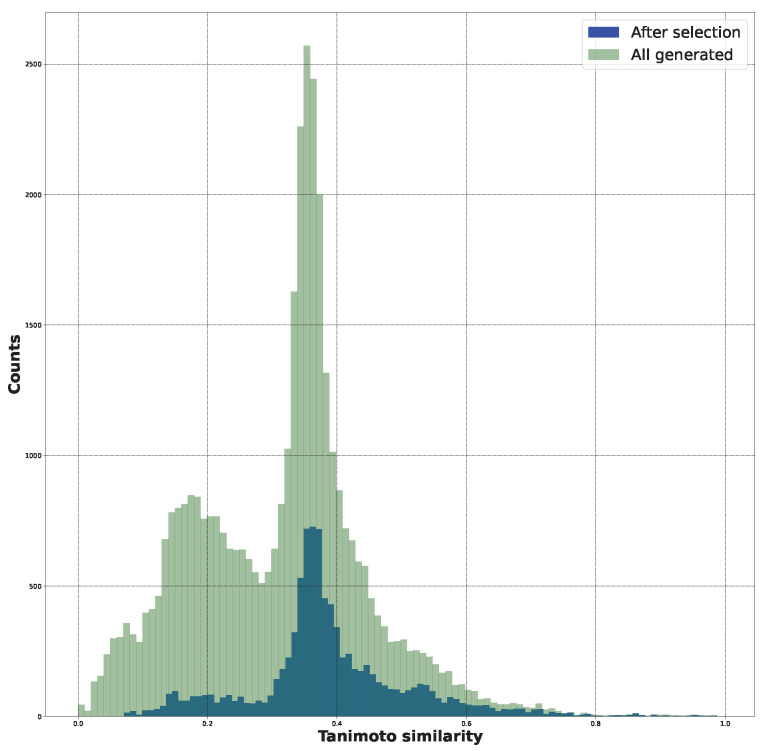
The Tanimoto similarity distributions for the initial structures, all the generated structures, and the SYBA-selected structures.

**Figure 4 ijms-24-11349-f004:**
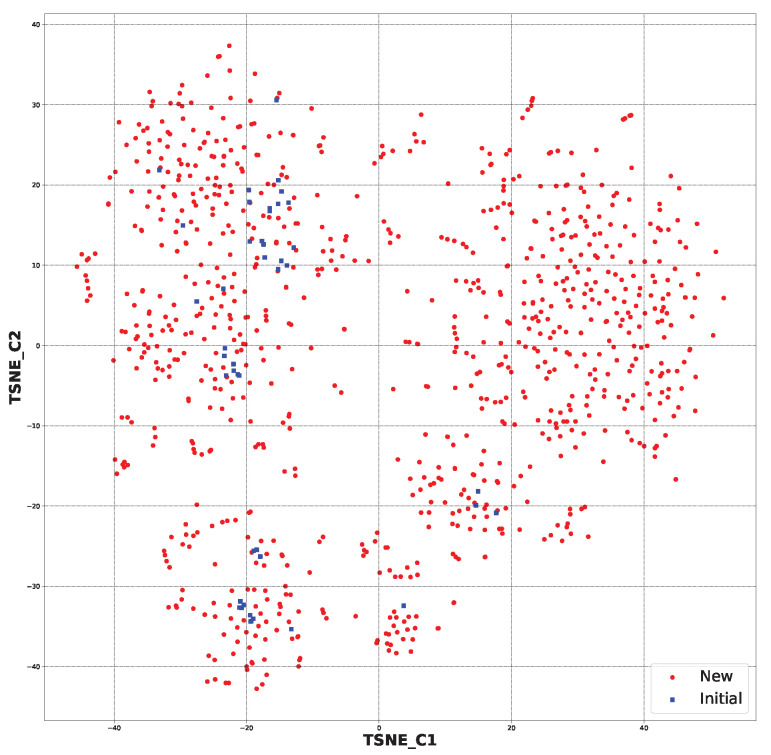
The chemical space for all the newly generated structures (891) and initial ones (44) based on molecular fingerprints.

**Figure 5 ijms-24-11349-f005:**
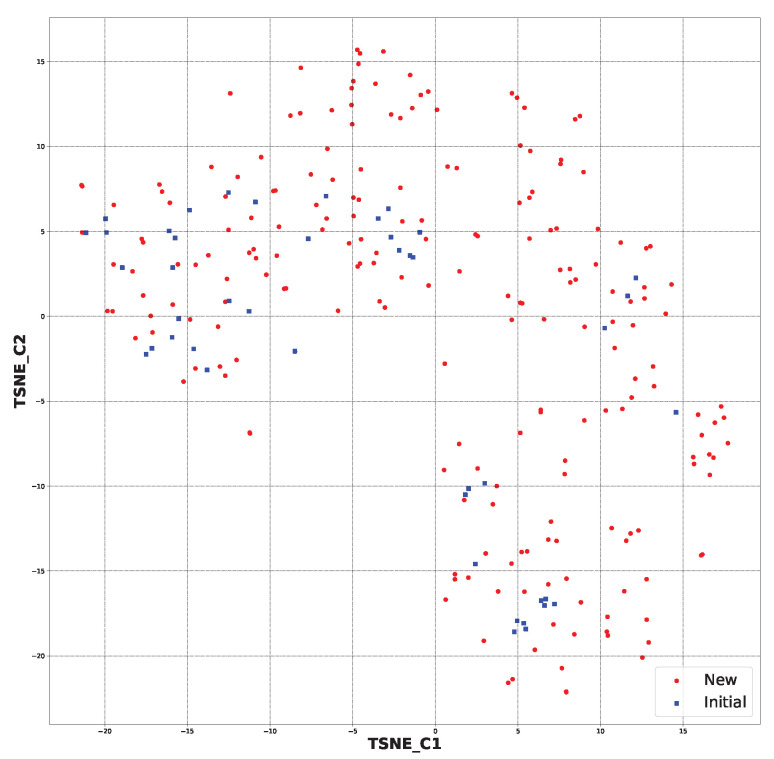
The chemical space for the SYBA-selected newly generated structures (213) and initial ones (44) based on molecular fingerprints.

**Table 1 ijms-24-11349-t001:** The structures of selected indole-based compounds. The table contains three initial structures with the assigned cytoprotective activities, Tanimoto similarity, and SMILES codes.

Initial structures’ images ^1^		
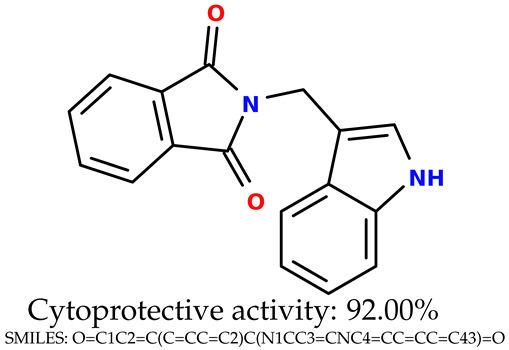	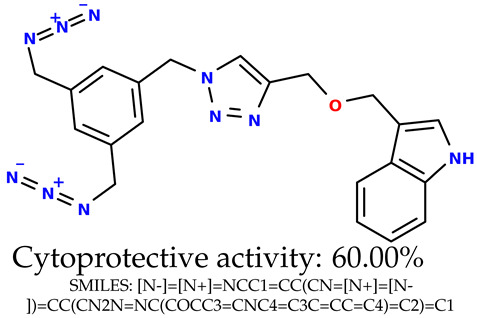	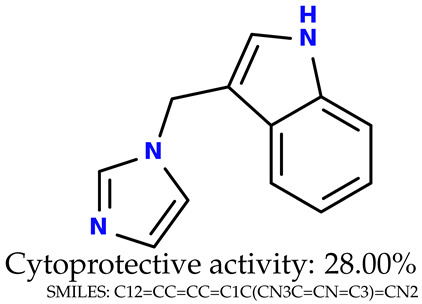
Newly generated structures’ images ^1^		
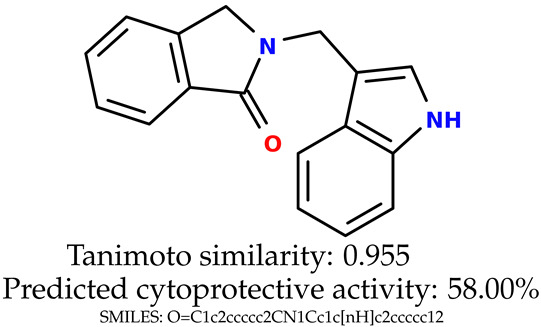 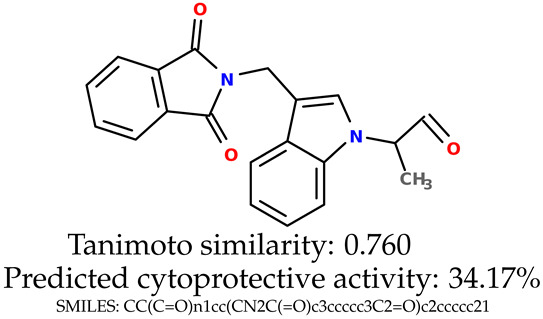 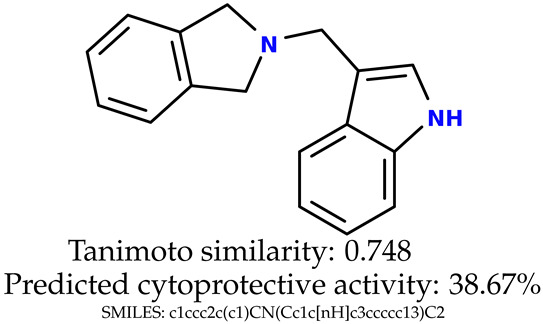	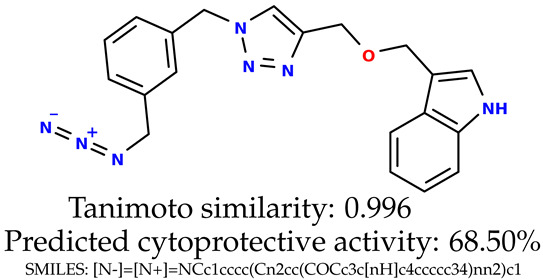 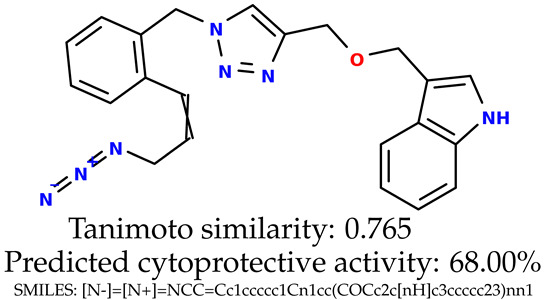 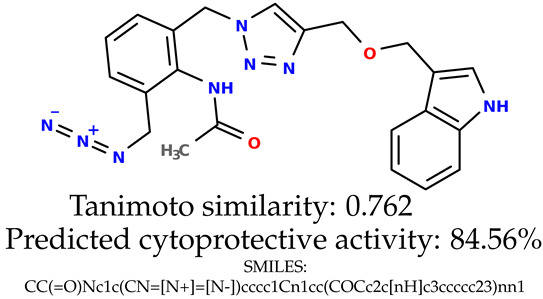	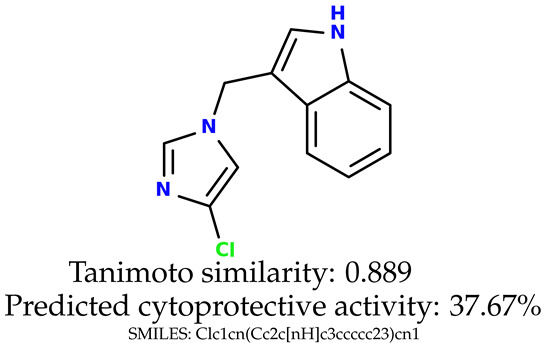 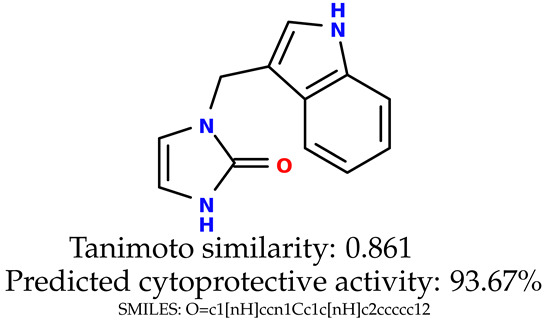 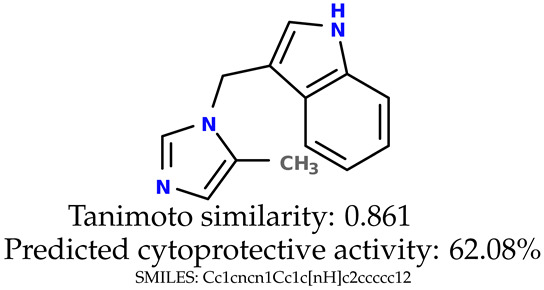

^1^ All the images were generated with the usage of Open Babel software version 3.1.1 [23,24].

**Table 3 ijms-24-11349-t003:** The indole-based structures for which cytoprotective activity was experimentally verified. The table contains information about predicted cytoprotective activity, measured cytoprotective activity, the highest Tanimoto similarity to the initial structures, and the SMILES code.

Tested structures’ images ^1^		
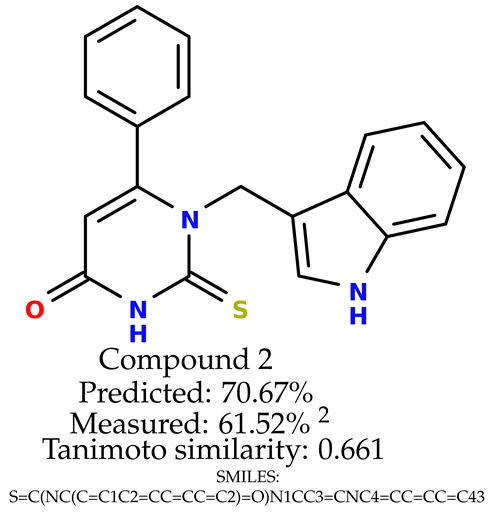	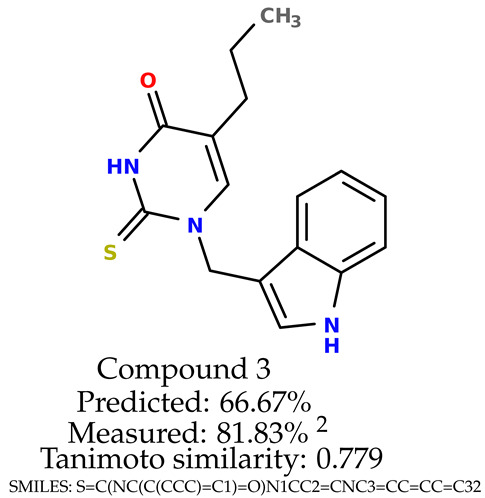	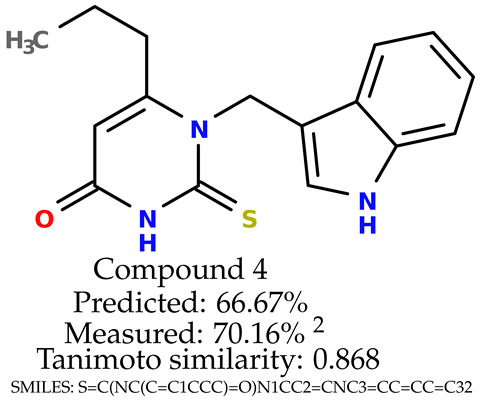

^1^ All the images were generated with the usage of Open Babel software version 3.1.1 [23,24]; ^2^ The measurements can be found in the Appendix A).

## Data Availability

All publication-related information can be accessed at this address: https://github.com/XDamianX-coder/Indole_new_structures (accessed on 12 June 2023). The Appendix A can also be downloaded using the link provided above.

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
