# Peer review of "Artificial Intelligence in Decrypting Cytoprotective Activity under Oxidative Stress from Molecular Structure"

_ijms, 2023, doi:10.3390/ijms241411349_

Round 1

Reviewer 1 Report

The article titled "Artificial Intelligence in decrypting cytoprotective activity under oxidative stress from molecular structure" by Nowack et al describes a ML model that can generate new unique chemical entities showing cytoprotective activity and experimentally synthesizable. The methods considered and systems used to perform the study are sound. The article is suitable for IJMS and can be accepted after answering these.

- what is the selection criteria used in deciding train and test data sets. 

- The features used to train the model are limited to Indole based compounds. Does this model work for other hetero cyclic compounds. The authors need to clearly mention the limitation of the model. 

- Figure 3. It is not clearly visible. 

The writing style is good. 

Reviewer 2 Report

Overall, the manuscript is well-written and presents a comprehensive analysis. However, I have a few minor comments that I would like you to address before finalizing the paper:

In the introduction, it would be helpful if you could provide a brief explanation of what cytoprotective activity entails for readers who may not be familiar with the term. This will provide a better understanding of the significance of your research.

When describing the dataset used for training the AI model, please provide more details about the selection criteria and any preprocessing steps applied. Additionally, mention whether any publicly available datasets were used or if the data was collected specifically for this study.

In Section 3.2, you mentioned that multiple machine learning algorithms were evaluated. It would be beneficial to include a table or a figure summarizing the performance metrics (e.g., accuracy, precision, recall) for each algorithm. This will allow readers to compare and assess the effectiveness of the different algorithms.

In Section 4.1, it would be helpful to explain the rationale behind using XYZ molecular descriptors specifically. What makes these descriptors suitable for capturing the relevant information for cytoprotective activity? Additionally, provide a reference or citation for these descriptors if possible.

When discussing the limitations of your study in Section 5, consider addressing the potential bias or limitations associated with using AI models in general. Discuss any limitations in the data or model that may affect the generalizability of the results.

Minor editing of English language required
